# A cross sectional study to examine factors influencing COVID-19 vaccine acceptance, hesitancy and refusal in urban and rural settings in Tamil Nadu, India

Ashish Joshi[1,2,3]*, Krishna Mohan Surapaneni[3], Mahima Kaur[3,4], Ashruti Bhatt[3,4], Denis Nash[1,2], Ayman El-Mohandes[1]

**1** City University of New York Graduate School of Public Health and Health Policy, New York, New York, United States of America, **2** CUNY Institute for Implementation Science in Population Health, New York, New York, United States of America, **3** SMAART PHIC, Panimalar Medical College Hospital & Research Institute, Chennai, India, **4** Foundation of Healthcare Technologies Society, Delhi, India

* ashish.joshi@sph.cuny.edu

**Data Availability Statement:** All relevant data are within the paper and its Supporting Information files.

## Abstract

The second wave of the COVID-19 pandemic left the Indian healthcare system overwhelmed. The severity of a third wave will depend on the success of the vaccination drive; however, even with a safe and effective COVID-19 vaccine, hesitancy can be an obstacle to achieving high levels of coverage. Our study aims to estimate the population's acceptance of the COVID-19 vaccine in an Indian district. A pilot community-based cross-sectional study was conducted from March-May 2021. The data was collected from eight primary health centres in Tamil Nadu. The eligible study participants were interviewed using a self-constructed questionnaire. A total of 3,130 individuals responded to the survey. Multinomial logistic regression was performed to assess the factors influencing COVID-19 vaccine hesitancy and refusal. Results of our study showed that 46% percent (n = 1432) of the respondents would accept the COVID-19 vaccine if available. Acceptance for the COVID-19 vaccine was higher among males (54%), individuals aged 18-24years (62%), those with higher education (77%), having the higher income (73%), and employed (51%). Individuals with no education (OR: 2.799, 95% CI = 1.103–7.108), and low income (OR: OR: 10.299, 95% CI: 4.879–21.741), were significant predictors of vaccine hesitancy (p < 0.05). Living in urban residence (OR: 0.699, 95% CI = 0.55–0.888) and age between 18 to 25 years (OR: 0.549, 95% CI = 0.309–0.977) were protective factor of COVID-19 vaccine hesitancy. While individuals in the age group 25-54years (OR = 1.601, 95%CI = 1.086–2.359), fewer education (OR = 4.8, 95% CI = 2.448–9.412,), low income (OR = 2.628, 95% CI = 1.777–3.887) and unemployment (OR = 1.351, 95% CI = 1.06–1.722) had high odds of refusing the COVID-19 vaccine. Concerns and suspicions about the safety of the COVID-19 vaccine (63%) was the major reasons causing hesitancy towards the COVID-19 vaccine The public health authorities and government need to design, develop and implement targeted interventions to enhance awareness about COVID-19 vaccines, and barriers and enablers to vaccine acceptance among individuals across diverse settings. Emphasis on involving local and religious leaders, ASHA workers, community healthcare workers, Anganwadi workers,

**Funding:** The author(s) received no specific funding for this work.

**Competing interests:** The authors have declared that no competing interests exist.

and auxiliary nurse midwives can help to overcome context-specific barriers in areas of low COVID-19 vaccine acceptance, especially in rural settings.

## Introduction

Vaccination is among the most significant public health interventions in reducing the spread and mortality caused by numerous infectious diseases [1]. As per World Health Organization (WHO) at best 10 million deaths were averted between 2010 and 2015 due to vaccinations [2]. The COVID-19 pandemic has caused a huge burden of morbidity and mortality, along with severe disruption of social and economical stability worldwide. One of the crucial strategies to eliminate this pandemic was the development of a safe and effective COVID-19 vaccine [3]. Even though, globally, the challenge of developing a safe, accessible, effective, and affordable COVID-19 vaccine has been overcome; vaccine hesitancy poses a threat to eliminate further outbreaks of this contagious infection [4].

Concerns about vaccine hesitancy are growing globally, especially in populous countries with low literacy. This prompted the WHO to declare it among the top ten health threats in 2019 [5]. In 2015, the WHO Strategic Advisory Group of Experts (SAGE) Working Group defined vaccine hesitancy as a "*delay in acceptance or refusal of vaccines despite availability of vaccinations services*" [6].

A vaccination program can help in achieving herd immunity but it necessitates coverage of a sufficient percentage of the population. The success of mass vaccination drives also depends on individuals' willingness to accept the vaccine. If large segments of the population are hesitant, herd immunity would be difficult to attain [7]. The findings of a European survey revealed that 73.9% of the 7664 participants from Denmark, France, Germany, Italy, Portugal, the Netherlands, and the UK would be willing to get the COVID-19 vaccine if available. A further 18.9% of respondents were not sure, and 7.2% did not want to get the COVID-19 vaccine [7]. In another survey conducted globally in June 2020 by Jeffrey Lazarus and colleagues, 742 respondents from India took part and it found that 44% and 30% of the respondents completely agree and somewhat agreed respectively to take the vaccine if available. About 14% were neutral, 5.66% somewhat disagreed and 5.66% of the respondents completely disagreed to receive the vaccine if available [8]. Several determinants such as historical, socioeconomic, cultural, ecological, health system/institutional, and political factors can influence whether an individual refuses, delays, or accepts vaccines [9].

The second wave of the COVID-19 pandemic hit India hard overwhelming the healthcare system nationally. Nearly six months after the peak of the first wave in September-October 2020, there was again an exponential increase in the daily cases. On April 15, 2021, the number of new cases was about two lakhs that is more than double the first peak [10]. In India, there has been increasing speculation on the possibility of a future third wave of infection, posing a burden on the healthcare system. It is speculated that a third wave may infect a smaller number of people and is unlikely to be severe. Nevertheless, the virus is still evolving and highly contagious variants of the virus may refute such speculations. The emergence of contagious and/or virulent variants, lower immunity of the vulnerable individuals, and lack of participation in COVID-19 preventive measures may contribute to a resurgence of the COVID-19 cases [11, 12].

By the 12[th] of September 2021, only 12.5% of the total Indian population was fully vaccinated [13]. There is insufficient data on the rates of COVID-19 vaccine acceptance and factors contributing to them in the Indian population.

The objectives of the study are to determine the factors influencing COVID-19 vaccine hesitancy and acceptance among urban and rural populations of India. To our knowledge, this pilot study is amongst the few to measure individuals' acceptance and hesitancy (delay or refusal) toward COVID-19 vaccines both in the urban and rural settings of India. Our findings will help design and field targeted interventions to facilitate uptake of the COVID-19 vaccines in India.

## Materials and methods

As per the National Family Health Survey (NFHS) 1 (1992–1993), in Tamil Nadu, the vaccine coverage including children aged 12–23 months (fully immunized with BCG, measles, and 3 doses each of polio and DPT) was 65%, which increased to 82% in the year 1998–1999 (NFHS 2). This was followed by 81% vaccine coverage in NFHS 3 (2005–2006) to a substantial decline to 69.7% as per the NFHS 4 (2015–16) in Tamil Nadu [14–16]. Even though Tamil Nadu has high literacy rates lower vaccine coverage is worrisome. In the early weeks of May 2021, Tamil Nadu stood 10th out of 37 States/UTs in India in administering total COVID-19 vaccine doses [16, 17]. Reduced vaccine coverage seen in the NFHS data, makes it necessary to assess the key predictors of COVID-19 vaccine hesitancy in Tamil Nadu with the possibility of a fourth wave in India.

The study was conducted at eight primary health centres (PHCs) in Tamil Nadu, India using the CO-VIN-CAP survey to assess the prevalence of COVID-19 vaccine acceptance and hesitancy, and its predictors among the adult [16].

- **Study Design, Duration, and Setting:** A pilot community-based cross-sectional study conducted from March 2021 to May 2021. The data was collected by interviewing eligible study participants at a one-time point using a self-constructed questionnaire (S1 Appendix). The study participants were recruited from selected urban and rural settings of Tamil Nadu, a southern state in India. Data were collected from eight PHCs namely, Thiruninravur, Thirumazhisai, Kathavur and Pallavedu, Mittanamallee, Vilinjiyamabakkam, Paruthipattu, Ulundai, and Poonamallee. It was made sure that all the selected PHCs were comparable with regard to available resources and living conditions. The participants visited PHCs to seek care and for general health check-ups.

- **Study Participants:** The study enrolled a total of 3,130 individuals for the pilot study using a non-probability complete enumeration sampling method [18]. Individuals aged 18 years plus, residing in the urban/rural settings of Tamil Nadu who gave consent to participate in the study were included in the study. Individuals having any mental or physical disability (accompanied by the caretaker), pregnant and lactating women and individuals who did not give consent were not included in the study.

- **Questionnaire and Variables gathered:** The CO-VIN-CAP survey was utilized to gather the data from the study participants [16]. We developed the survey instrument and the significant items identified based on prior literature findings were included in our survey instrument to adapt to our study context [8, 19–24]. Variables assessed include:

  ○ *Socio-demographics*: The data collected included participants' age, gender, income level [25], education level [25], employment status, occupation [25], region, religion, marital status, and. parenthood status.

  ○ *Health status profile*: The data was recorded on co-morbidities, COVID-19 diagnosis, and anthropometry measurements of the respondents. BMI is calculated as weight in kilograms divided by height in meters squared ($kg/m^2$). Standard BMI classification was

applied to categorize respondents: underweight (BMI < 18.5), normal weight (BMI 18.5–24.9), overweight (BMI 25.0–29.9), and obese (BMI ≥ 30) [26].

○ *Prior immunization* and *COVID-19 vaccine status* Information was collected related to prior experiences of immunizations and the COVID-19 vaccine status. Data on prior refusal to other vaccines, any adverse reaction experienced due to the vaccine, and any event leading to discouragement in taking a COVID-19 vaccine was gathered from the respondents. Information on the COVID-19 vaccine status from the respondents was gathered by the following question: Have you received a COVID-19 vaccine? Response questions included: Received one dose of vaccine, received two doses of vaccine, and No dose of vaccine received. Further, a follow-up question was asked about the plan to receive a second dose of the COVID-19 vaccine from the respondent who received the first dose. Additionally, data was gathered for the reasons to refuse the second dose of the COVID-19 vaccine as an open-ended question.

○ *History of COVID-19 disease*: Data was gathered regarding individual immediate and close family members' history and severity of COVID-19 infection.

○ *COVID-19 vaccine acceptance and hesitancy*: Information was recorded from the respondents about their preferences to accept/delay/ completely reject the administration of the COVID-19 vaccine.

➢ *Vaccine Acceptance and vaccine refusal*: Vaccine acceptance is defined as the individual or group's decision to accept or refuse when presented with an opportunity to vaccinate [27]. Participants were asked about the willingness to accept or reject the vaccine if made available in general as well as if it is given free of cost. General acceptance and refusal of the COVID-19 vaccine by the respondents measured by the following question: Would you accept the COVID-19 vaccine if available? Response options included "Yes", "Maybe yes", "Not sure", "Maybe not", and "No". Acceptance of COVID-19 vaccine if made available was for the outcome assessment. Data regarding the acceptance of the COVID-19 vaccine with different levels of efficacy and effectiveness (50%, 70%, 90–95%) was also gathered. The general acceptance of a COVID-19 vaccine was our primary outcome variable, which dichotomized as acceptance ("definitely yes, will accept the COVID-19 vaccine") and refusal ("definitely no, will not accept COVID-19 vaccine"). Data on reasons for accepting or refusing a COVID-19 vaccine were also gathered.

➢ *Vaccine Hesitancy*: In 2015, the WHO Strategic Advisory Group of Experts (SAGE) Working Group on Vaccine Hesitancy defined vaccine hesitancy as a "delay in acceptance or refusal of vaccines despite availability of vaccinations services. Vaccine hesitancy is complex and context-specific, varying across time, place, and vaccines. It is influenced by factors such as complacency, convenience, and confidence" [6]. In this study any respondent who answered Maybe yes", "Not sure", "Maybe not", to the question: Would you accept COVID-19 vaccine available, were considered hesitant towards getting a COVID-19 vaccine. Respondents were asked to specify the reason behind their hesitancy toward taking a COVID-19 vaccine. Additionally, the survey gathered information about things that would motivate individuals to get the vaccine and reduce their hesitancy.

The Institutional ethical clearance was approved from the Panimalar Medical College Hospital & Research Institute-Institutional Human Ethics Committee (PMCHRI-IHEC): CDSCO Registration No. ECR/1399/Inst/TN/2020 with approval Number PMCH&RI/IHEC/2021/037

dated 13.01.2021. The study was conducted according to the Declaration of Helsinki as the current study involves human subjects [28]. The IRB-approved informed consent form was administered in English and local language by the research team to the eligible individuals. A written informed consent was obtained from the respondents to participate in the study (S2 Appendix). The research team described the study, the time required, and the benefits of the study result to the participants. Those who gave the consent were enrolled in the study. The study protocol was published in July 2021 [16].

## Statistical analysis

Descriptive analysis was conducted to report the mean and percentages and frequency for the continuous and categorical variables respectively. Stratified analysis was conducted to analyze the impact of various factors across the distribution of region of residence. Acceptance of the COVID-19 vaccine if available was the main outcome measure of the study. Multinomial logistic regression was performed to determine the predictors that influence the acceptance of the COVID-19 vaccine among the individuals in Tamil Nadu. To conduct the analysis, the outcome was trichotomized as acceptance, hesitant (maybe yes, not sure, maybe no), and refusal. The individuals who were fully vaccinated were characterized into the acceptance group. All analysis was performed using SPSS v.24 Result findings of the study were presented as crude/unadjusted odds ratios (OR) with 95% confidence intervals and a p-value less than 0.05 was considered significant.

## Results

### Population characteristics

Data on 3,130 individuals gathered (S3 Appendix). The majority of the study participants were in the age group 25–54 years (50%, n = 1540); 52% (n = 1574) were females, (24%, n = 743) had high school, (61%, n = 1874) were married and more than half of them (57%, n = 1799) were residing in urban areas. Most of the respondents were employed or self-employed (61%, n = 1916) Table 1.

### Health status

The majority of the respondents did not have any underlying physician confirmed illness (73%, n = 2240). Among the respondents who had physician confirmed illness, More than half of the respondents (69%; n = 2127) were not currently enrolled in any health insurance program. More than half of the respondents had been tested for COVID-19 (55%; n = 1667), and for those who were tested positive for COVID-19, the majority had no symptoms (n = 51,30%). Of those after positive test results, the majority practiced home remedies to treat the infection (45%; n = 71). S4 Appendix gives in detail the health status of the respondents.

### Prior vaccine and COVID-19 vaccination status

Data related to prior history of immunizations gathered (Table 2). More than half of the individuals (65%, n = 2005) had never reported refusing to take any vaccine. Seventy-six percent (n = 2,302) of the individuals did not report any kind of bad reaction ever to any vaccine administration. Fifty-nine percent (n = 1807) of them reported that vaccination-related bad events will not discourage them from taking the COVID-19 vaccine. One-fourth of the individuals reported taking at least one dose of the COVID-19 vaccine while 19% (n = 565) reported receiving both doses of the COVID-19 vaccine. More than half of the respondents had not received the COVID-19 vaccine (56%, n = 1686). Of those individuals who had

**Table 1. Socio-demographic characteristics of the study sample.**

| Socio-Demographics | Attributes | Total, n (%) | Urban, n (%) | Rural, n (%) |
|---|---|---|---|---|
| Gender (n = 3011) | Male | 1437 (48) | 879 (48) | 553 (48) |
| | Female | 1574 (52) | 967 (52) | 601 (52) |
| Age (n = 3088) | 18–24 | 634 (21) | 414 (22) | 216 (18) |
| | 25–54 | 1540 (50) | 993 (52) | 543 (46) |
| | 55–64 | 691 (22) | 386 (20) | 302 (26) |
| | 65+ | 223 (7) | 108 (6) | 115 (10) |
| Education level (n = 3080) | No education | 433 (14) | 201 (11) | 231 (20) |
| | Primary school | 242 (8) | 140 (7) | 101 (9) |
| | Middle School | 345 (11) | 191 (10) | 151 (13) |
| | High school | 743 (24) | 470 (25) | 272 (23) |
| | Intermediate or diploma | 584 (19) | 396 (21) | 187 (16) |
| | Graduate | 616 (20) | 410 (22) | 201 (17) |
| | Profession or Honours | 117 (4) | 91 (5) | 26 (2) |
| Monthly Income (INR) (n = 2737) | ≤10,001 | 410 (15) | 245 (15) | 163 (15) |
| | 10,002–29,972 | 319 (12) | 199 (12) | 116 (11) |
| | 29,973–74,755 | 681 (25) | 413 (25) | 266 (24) |
| | 74,756–99,930 | 631 (23) | 349 (21) | 282 (26) |
| | 99,931–199,861 | 316 (12) | 170 (10) | 146 (13) |
| | ≥199,862 | 380 (14) | 255 (16) | 124 (11) |
| Marital status (n = 3089) | Unmarried | 1045 (34) | 698 (37) | 347 (29) |
| | Married | 1874 (61) | 1126 (59) | 748 (63) |
| | Others | 170 (6) | 79 (4) | 91 (8) |
| Employment Status (n = 3035) | Employed | 1916 (63) | 1154 (62) | 762 (66) |
| | Homemaker | 239 (8) | 160 (9) | 79 (7) |
| | Others | 880 (29) | 561 (30) | 319 (28) |

received the first dose of the COVID-19 vaccine, majority (87%, n = 684) of them indicated they would take the second dose.

**Vaccine acceptance and hesitancy.** Forty-six percent (n = 1432) of the respondents would accept the COVID-19 vaccine if available. About one-fourth (20%; n = 611) of the respondents were inclined to accept the COVID-19 vaccine. Nearly a quarter of the individuals (25%; n = 766) were not sure if they would take the COVID-19 vaccine if made available. In totality, 46% (n = 874) of the respondents living in the urban area were hesitant towards getting the COVID-19 vaccine whereas people living in rural areas were more hesitant to get vaccinated with COVID-19 vaccine (55%; n = 629) (Table 3).

The effectiveness of the vaccine was reported as an important factor to enhance COVID-19 vaccination uptake. Around 32% (n = 991) of the individuals would accept a COVID-19 vaccine with 90–95% effectiveness compared to 25% (n = 759) if the COVID-19 vaccine had 70% effectiveness and 23% (n = 705) if the COVID-19 vaccine is 50% effective (Table 3).

The main reason for individuals to get a COVID-19 vaccine dose included safety from the COVID-19 infection (77%) (Fig 1). Concerns and suspicions about the safety of the COVID-19 vaccine (63%) and the fear from the COVID-19 vaccine (17%) were the major reasons causing hesitancy towards the COVID-19 vaccine (Fig 2). The factors that would motivate to receive the COVID-19 vaccine would include the recommendation from the health professional (35%), increased awareness about the COVID-19 vaccine (12%), and protection of family members from COVID-19 infection (10%) (Fig 3).

**Table 2. COVID-19 vaccination and prior vaccination status of the respondents.**

| Variables | Attributes | Total n (%) | Urban n (%) | Rural n (%) |
|---|---|---|---|---|
| In the past, have you ever refused to take a vaccine? (n = 3076) | Yes | 941 (31) | 527 (28) | 410 (35) |
| | No | 2005 (65) | 1285 (68) | 715 (61) |
| | Not willing to tell | 130 (4) | 73 (4) | 56 (5) |
| In the past, have you ever had a bad reaction to a vaccine? (n = 3019) | Yes | 430 (14) | 248 (13) | 180 (16) |
| | No | 2302 (76) | 1453 (79) | 843 (73) |
| | Not sure | 216 (7) | 117 (6) | 96 (8) |
| | Not willing to tell | 71 (2) | 31 (2) | 40 (3) |
| Do you know anyone who has had a bad reaction to a vaccine previously? (n = 3065) | Yes | 680 (22) | 393 (21) | 284 (24) |
| | No | 1732 (57) | 1018 (54) | 710 (60) |
| | Not sure | 521 (17) | 350 (19) | 168 (14) |
| | Not willing to tell | 132 (4) | 116 (6) | 16 (1) |
| Would the vaccination related bad events discourage you from taking a COVID-19 vaccine? (n = 3071) | Yes | 355 (12) | 201 (11) | 150 (13) |
| | No | 1807 (59) | 1068 (57) | 735 (62) |
| | Not sure | 720 (23) | 488 (26) | 230 (20) |
| | Not willing to tell | 189 (6) | 125 (7) | 63 (5) |
| Have you received a COVID-19 vaccine? (n = 3006) | Received one dose of vaccine | 758 (25) | 443 (24) | 315 (27) |
| | Received two doses of vaccine | 565 (19) | 369 (20) | 196 (17) |
| | No | 1683 (56) | 1037 (56) | 646 (56) |
| If you received first of COVID-19 vaccine, do you plan to take the second dose of it? (n = 784) | Yes | 684 (87) | 398 (88) | 286 (87) |
| | No | 58 (7) | 32 (7) | 26 (8) |
| | Not sure | 42 (5) | 24 (5) | 18 (5) |
| If No, please specify the reason (n = 64) | Age | 33 (52) | 7 (37) | 26 (57) |
| | Adverse reaction | 4 (6) | 1 (5) | 3 (7) |
| | Medical conditions | 5 (8) | 2 (11) | 3 (7) |
| | Others | 23 (34) | 9 (47) | 14 (30) |

**Association between socio-demographics and COVID-19 vaccine acceptance.** Table 4 describes the association between socio-demographic factors and the acceptance of the COVID-19 vaccine. COVID-19 vaccine acceptance was significantly higher among males (54%), individuals in the age group 18–24 years (62%), those with professional education levels (77%), and among respondents in higher income levels (73%). In addition, those working and were employed had higher acceptance of the COVID-19 vaccine (51%). The percentage of vaccine acceptance (55%) was also higher among respondents residing in urban settings. Age (p<0.001), education (p<0.001), monthly income (p < 0.001), region of residence (p = 0.014), marital status (p = 0.038) and employment status (p = 0.036), were significantly associated with the acceptance of COVID-19 vaccine.

COVID-19 vaccine acceptance was higher among individuals enrolled in private insurance plans (64%). Additionally, COVID-19 vaccine acceptance was higher among respondents who experienced moderate symptoms and sought help from a doctor (81%) (S5 Appendix).

Table 5 demonstrates the association of Prior vaccination and COVID-19 vaccine status vs. acceptance of COVID-19 vaccine.

## Predictors of COVID-19 vaccine acceptance

**COVID-19 vaccine acceptance: Not sure (hesitant) V/s yes (acceptance).** No education (OR = 2.799, 95% CI = 1.103–7.108, p = 0.03) and individuals with lower levels of income are

**Table 3. COVID-19 vaccine acceptance and hesitancy among the respondents.**

| Variables | | Yes n(%) | May be yes n (%) | Not sure n (%) | May be not n (%) | No n(%) |
|---|---|---|---|---|---|---|
| Would you accept a COVID-19 vaccine if available? (n = 3073) | Total | 1432 (46) | 611 (20) | 766 (25) | 148 (5) | 150 (5) |
| | Urban | 929 (49) | 330 (17) | 464 (24) | 80 (4) | 91 (5) |
| | Rural | 490 (41) | 263 (22) | 299 (25) | 68 (6) | 59 (5) |
| Would you accept a COVID-19 vaccine if available for free? (n = 3081) | Total | 1076 (32) | 558 (16) | 867 (26) | 374 (11) | 240 (7) |
| | Urban | 704 (37) | 322 (17) | 471 (25) | 259 (14) | 144 (8) |
| | Rural | 360 (30) | 224 (19) | 391 (33) | 111 (9) | 95 (8) |
| Would you accept a COVID-19 vaccine if it is 50% effective? (n = 3115) | Total | 707 (23) | 636 (20) | 970 (31) | 308 (10) | 494 (16) |
| | Urban | 429 (23) | 380 (20) | 535 (28) | 206 (11) | 349 (18) |
| | Rural | 261 (22) | 245 (21) | 431 (36) | 101 (9) | 144 (12) |
| Would you accept a COVID-19 vaccine if it is 70% effective? (n = 3097) | Total | 759 (25) | 660 (21) | 840 (27) | 319 (10) | 519 (17) |
| | Urban | 484 (26) | 384 (20) | 455 (24) | 207 (11) | 359 (19) |
| | Rural | 257 (22) | 265 (23) | 385 (33) | 110 (9) | 158 (13) |
| Would you accept a COVID-19 vaccine with 90–95% effective? (n = 3098) | Total | 991 (32) | 675 (22) | 895 (29) | 296 (10) | 241 (8) |
| | Urban | 676 (36) | 392 (21) | 483 (26) | 179 (9) | 160 (8) |
| | Rural | 299 (25) | 276 (24) | 406 (35) | 113 (10) | 80 (7) |
| Would you accept a COVID-19 vaccine if your employer recommended it? (n = 2988) | Total | 1298 (43) | 805 (27) | 615 (21) | 147 (5) | 123 (4) |
| | Urban | 793 (44) | 479 (26) | 393 (22) | 77 (4) | 71 (4) |
| | Rural | 495 (43) | 311 (27) | 217 (19) | 70 (6) | 48 (4) |
| Would you accept a COVID-19 vaccine if your doctor recommended it? (n = 3106) | Total | 1210 (39) | 637 (21) | 792 (25) | 287 (9) | 180 (6) |
| | Urban | 710 (37) | 372 (20) | 477 (25) | 206 (11) | 129 (7) |
| | Rural | 492 (42) | 257 (22) | 306 (26) | 75 (6) | 48 (4) |
| Would you accept a COVID-19 vaccine for children when it is available? (if applicable) (n = 2807) | Total | 470 (17) | 689 (25) | 731 (26) | 188 (7) | 729 (26) |
| | Urban | 274 (17) | 384 (23) | 465 (28) | 138 (8) | 396 (24) |
| | Rural | 177 (16) | 300 (27) | 263 (24) | 48 (4) | 329 (29) |
| Would you accept a COVID-19 vaccine if successfully developed and approved? (n = 3093) | Total | 834 (27) | 842 (27) | 738 (24) | 199 (6) | 480 (16) |
| | Urban | 555 (29) | 496 (26) | 424 (22) | 126 (7) | 286 (15) |
| | Rural | 265 (23) | 328 (28) | 313 (27) | 73 (6) | 193 (16) |

more likely to be unsure or hesitant towards the COVID-19 vaccine. For example, those whose salary ranged from INR. 10,002-INR 29,972/month were seven times more likely to be hesitant towards the COVID-19 vaccine than the individuals with a monthly salary of INR ≥199,862 (OR: 10.299, 95% CI: 4.879–21.741, p< 0.001). Living in urban residence (OR: 0.699, 95% CI = 0.55–0.888, p = 0.003) and age between 18 to 25 years (OR: 0.549, 95% CI = 0.309–0.977, p = 0.041) was protective factor of COVID-19 vaccine hesitancy (Table 6).

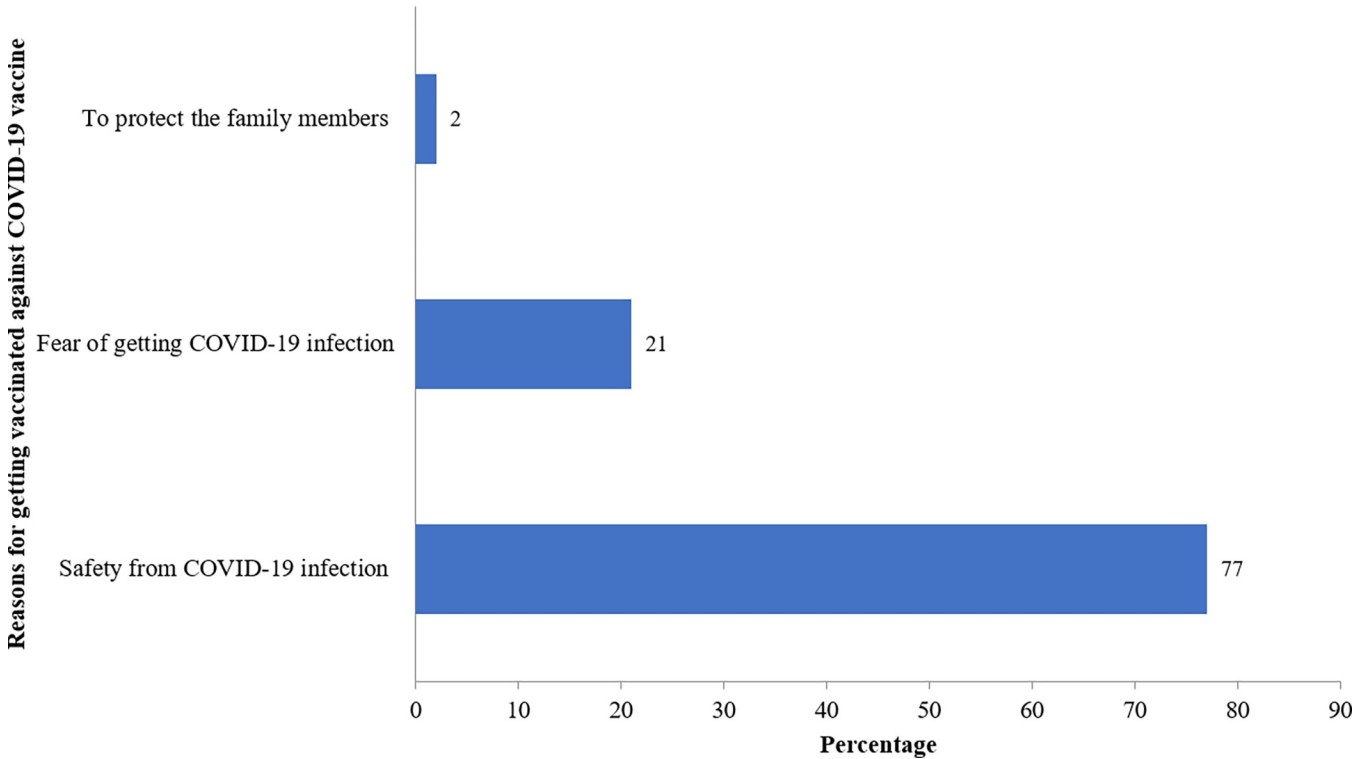

**Fig 1. Reasons for accepting a COVID-19 vaccine (n = 616).**

**COVID-19 vaccine acceptance: No (refusal) V/s yes (acceptance).** Individuals in the age group 25-54years are 1.6 times more likely to refuse the COVID-19 vaccine than the individuals in the age group 65 years and above (OR = 1.601, 95%CI = 1.086–2.359, p = 0.017). The

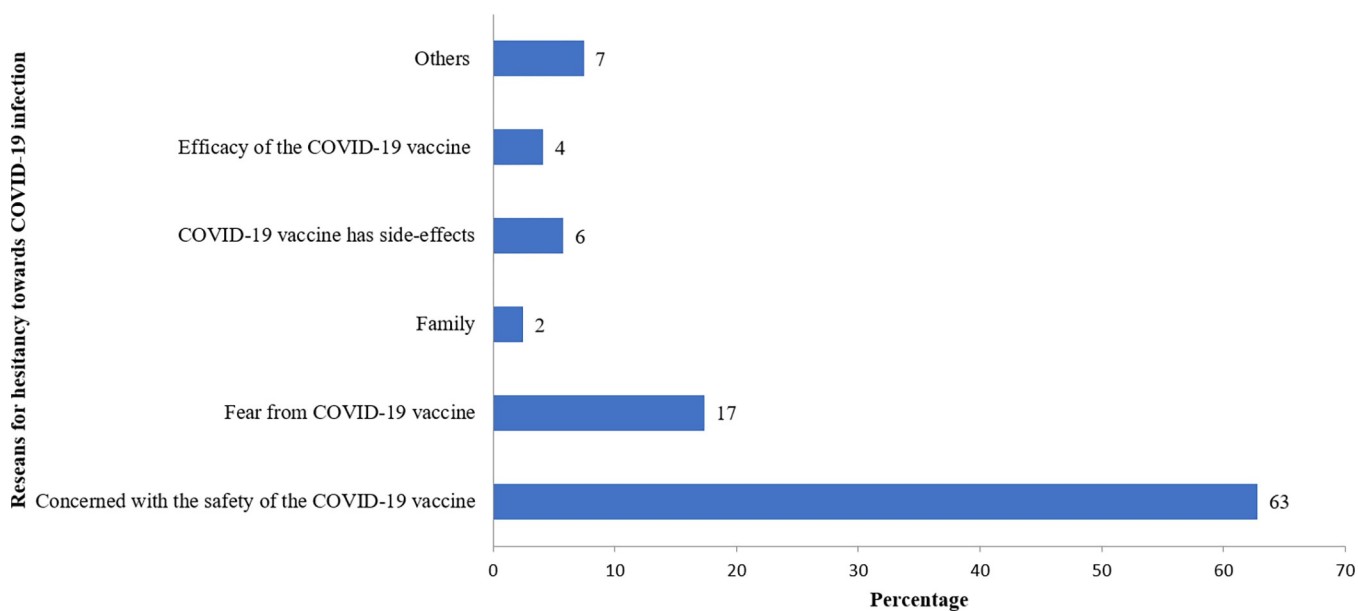

**Fig 2. Reasons leading to hesitancy towards COVID-19 vaccine (n = 121).**

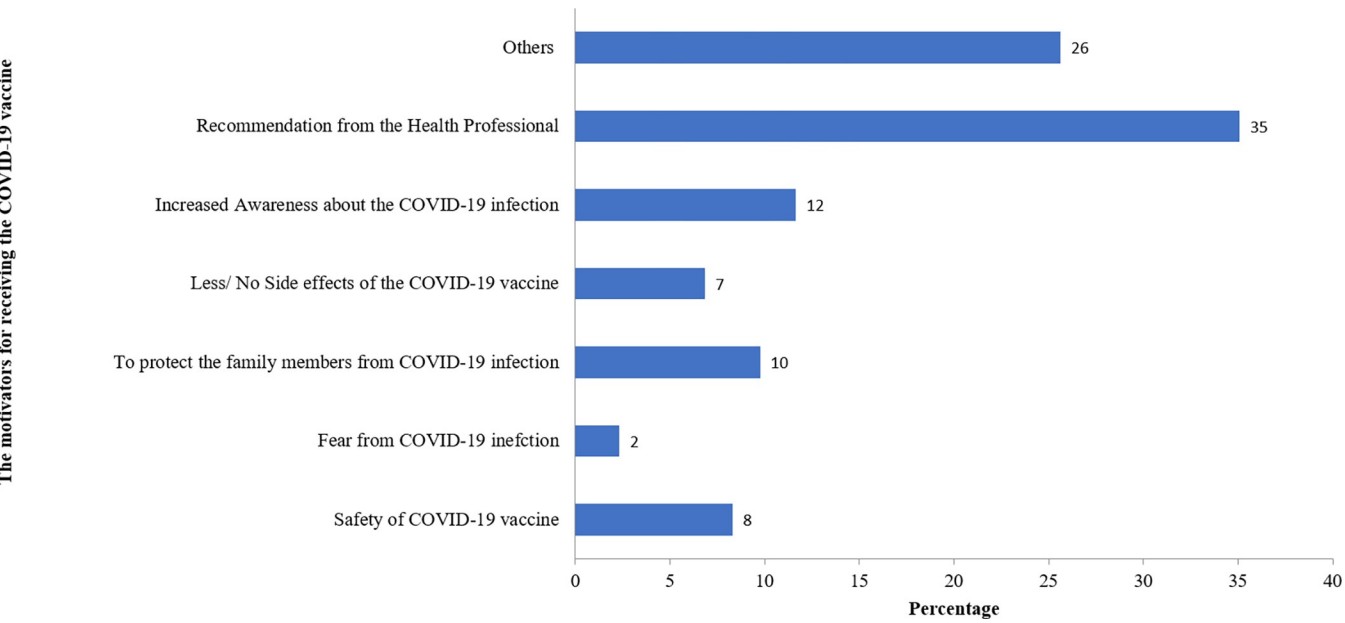

**Fig 3. Motivators behind receiving a COVID-19 vaccine (n = 687).**

odds of refusing the COVID-19 vaccine were 4.8 times higher among individuals with primary level education than individuals with the highest level of education (OR = 4.8, 95% CI = 2.448–9.412, p < 0.001). Those whose salary ranged from INR. 10,002-INR 29,972/month were 2.6 times more likely to refuse the COVID-19 vaccine than the individuals with monthly salary of INR ≥199,862 (OR = 2.628, 95% CI = 1.777–3.887, p< 0.001). Unemployed individuals, retired or lost jobs due to COVID-19 were more likely to refuse to take the COVID-19 vaccine as compared to employed individuals (OR = 1.351, 95% CI = 1.06–1.722, p = 0.015) (Table 6).

## Discussion

Lower vaccine acceptance rates and hesitancy towards vaccines is a major public health challenge as it can result in further outbreaks with deaths and orphanhood, as well as the emergence and spread of new variants that are not deterred by immunity conferred by existing vaccines and by prior COVID-19 infection [29, 30].

Results of our study showed that 46% percent (n = 1432) of the respondents would accept the COVID-19 vaccine if available. About one-fourth (20%; n = 611) of the respondents were inclined to accept the COVID-19 vaccine. In a global survey conducted, it was reported that 75% of the respondents in India would accept the COVID-19 vaccine which was higher than the acceptance reported in our study [8]. The acceptance rate was lower than the acceptability rate in Latin America. Brazil [8], Ecuador [31] and Mexico [8] reported more than 70% acceptance towards the COVID-19 vaccine. The COVID-19 vaccine acceptance rate in European countries 62% in France, 70% in Germany, 73% in the Netherlands, 74% in Italy (excluding Lombardy), 75% in Portugal, 79% in the United Kingdom, and 80% in Denmark) was higher than reported in India [7]. Similarly, in the Middle East, in UAE, and Saudi Arabia, the COVID-19 vaccine acceptance rate was 55% and 65% respectively [22, 32]. In Bangladesh, 75% vaccine acceptance was reported among the general population [33]. While caution is warranted in comparisons between studies as they were conducted at a different time, and were utilizing different methodologies to assess the rate of COVID-19 vaccine acceptance leading to variation.

**Table 4. Distribution of socio-demographic variables vs. acceptance of COVID-19 vaccine.**

| Variables | Attributes | Vaccine Acceptance n (%) | Vaccine Hesitant n (%) | Vaccine Rejection n (%) | Likelihood Ratio Chi-Square (p-value) |
|---|---|---|---|---|---|
| Gender (n = 2992) | Male (n = 1427) | 776(54) | 229(16) | 422(30) | $\chi 2 = 3.392\ (0.183)$ |
| | Female (n = 1565) | 816(52) | 263(17) | 486(31) | |
| Age (n = 3068) | 18–24 (n = 629) | 391(62) | 85(14) | 153(24) | **$\chi 2 = 24.822\ (<0.001)$** |
| | 25–54 (n = 1528) | 793(52) | 231(15) | 504(33) | |
| | 55–64 (n = 689) | 322(47) | 148(21) | 219(32) | |
| | 65+ (n = 222) | 114(51) | 38(17) | 70(32) | |
| Education level (n = 3060) | No education (n = 433) | 173(40) | 119(27) | 141(33) | **$\chi 2 = 78.608\ (<0.001)$** |
| | Primary school (n = 240) | 92(38) | 52(22) | 96(40) | |
| | Middle School (n = 341) | 152(45) | 59(17) | 130(38) | |
| | High school (n = 739) | 356(48) | 129(17) | 254(34) | |
| | Intermediate or diploma (n = 581) | 340(59) | 60(10) | 181(31) | |
| | Graduate (n = 611) | 409(67) | 76(12) | 126(21) | |
| | Profession or Honours (n = 115) | 89(77) | 7(6) | 19(17) | |
| Monthly Income (INR) (n = 2718) | ≤10,001 (n = 407) | 230(57) | 37(9) | 140(34) | **$\chi 2 = 168.648\ (<0.001)$** |
| | 10,002–29,972 (n = 313) | 127(41) | 50(16) | 136(43) | |
| | 29,973–74,755 (n = 680) | 316(46) | 150(22) | 214(31) | |
| | 74,756–99,930 (n = 630) | 364(58) | 151(24) | 115(18) | |
| | 99,931–199,861 (n = 313) | 201(64) | 25(8) | 87(28) | |
| | ≥199,862 (n = 375) | 272(73) | 12(3) | 91(24) | |
| Region of residence (n = 3076) | Urban (n = 1897) | 1037(55) | 277(15) | 583(31) | **$\chi 2 = 8.597\ (0.014)$** |
| | Rural (n = 1179) | 584(50) | 227(19) | 368(31) | |
| Marital status (n = 2281) | Unmarried (n = 1044) | 607(58) | 173(17) | 264(25) | **$\chi 2 = 10.148\ (0.038)$** |
| | Others (n = 172) | 84(49) | 26(15) | 62(36) | |
| | Married (n = 1065) | 938(50) | 304(16) | 623(33) | |
| Employment Status (n = 3024) | Others (n = 875) | 435(50) | 129(15) | 311(36) | **$\chi 2 = 10.301\ (0.036)$** |
| | Homemaker (n = 239) | 83(35) | 38(16) | 118(49) | |
| | Employed (n = 1910) | 1074(56) | 329(17) | 507(27) | |

Significance taken at $p < 0.05$ (In Bold)

Thirty-two percent of the individuals would accept a COVID-19 vaccine with 90–95% effectiveness compared to 23% if the COVID-19 vaccine is 50% effective. Similarly, previous studies conducted in Indonesia [21], Libya [34], Kuwait [3], have demonstrated that acceptance towards 95% effective COVID-19 vaccine was highest, and it declined with the decrease in the efficacy of the COVID-19 vaccine. Safety from the COVID-19 infection (77%) was the main reason stated by the individuals to get the COVID-19 vaccine.

In total, 46% of the respondents had acceptance towards the COVID-19 vaccine. In totality, a lower percentage (46%, n = 874) of the respondents living in the urban area were hesitant towards getting the COVID-19 vaccine in comparison to respondents in rural regions (55%; n = 629). This finding is striking bearing in mind that the pilot survey was conducted during March-April 2021 when India was witnessing the devastating second wave of the COVID-19 pandemic. About 5% of the respondents refused to be vaccinated, which is similar to the rejection rate in the UK and Turkey [23]. Our results show 42% of the respondents in the rural setting were more accepting of the COVID-19 vaccine if recommended by the doctors in comparison to the 37% in the urban setting. Literature has demonstrated that

**Table 5. Prior vaccination and COVID-19 vaccine status vs. acceptance of COVID-19 vaccine.**

| Variables | Attributes | Vaccine Acceptance n (%) | Vaccine Hesitant n (%) | Vaccine Rejection n (%) | Likelihood Ratio Chi-Square (p-value) |
|---|---|---|---|---|---|
| In the past, have you ever refused to take a vaccine? (n = 3057) | Yes (n = 939) | 507(54) | 251(27) | 181(19) | **22.076 (<0.001)** |
| | No (n = 1989) | 1040(52) | 237(12) | 712(36) | |
| | Not willing to tell (n = 129) | 62(48) | 12(9) | 55(43) | |
| In the past, have you ever had a bad reaction to a vaccine? (n = 3001) | Yes (n = 429) | 254(59) | 117(27) | 58(14) | 11.514 (0.074) |
| | No (n = 2288) | 1179(52) | 340(15) | 769(34) | |
| | Not sure (n = 214) | 117(55) | 33(15) | 64(30) | |
| | Not willing to tell (n = 70) | 29(41) | 4(6) | 37(53) | |
| Would the vaccination related bad events discourage you from taking a COVID-19 vaccine? (n = 3053) | Yes (n = 353) | 216(61) | 93(26) | 44(12) | **34.615 (<0.001)** |
| | No (n = 1796) | 1069(60) | 235(13) | 492(27) | |
| | Not sure (n = 717) | 277(39) | 137(19) | 303(42) | |
| | Not willing to tell (n = 187) | 43(23) | 35(19) | 109(58) | |
| Have you received a COVID-19 vaccine? (n = 2998) | Received one dose of vaccine (n = 753) | 491(65) | 128(17) | 134(18) | |
| | Received two doses of vaccine (n = 569) | 569(100) | 0(0) | 0(0) | |
| | No (n = 1676) | 527(31) | 362(22) | 787(47) | |
| If you received first COVID-19 vaccine, do you plan to take the second dose of it? (n = 779) | Yes (n = 678) | 455(67) | 104(15) | 119(18) | 3.107 (0.54) |
| | No (n = 58) | 26(45) | 17(29) | 15(26) | |
| | Not sure (n = 43) | 15(35) | 10(23) | 18(42) | |

Significance taken at p < 0.05 (In Bold)

recommendations from doctors could play a huge role in increasing the probability of acceptance of the COVID-19 vaccine among the general population [24, 35]. Besides, individuals living in rural settings hold a high level of trust among healthcare providers and doctors to deliver accurate, and reliable COVID-19 and its vaccine-related information. Engaging healthcare providers and doctors in addressing any medical mistrust and misinformation will build the trust and confidence of individuals in the COVID-19 vaccine and can play a crucial role in overcoming context-specific barriers in areas of low vaccine acceptance and design effective strategies to enhance vaccine acceptance.

Among the urban settings, a higher percentage (44%) of the respondents would accept a COVID-19 vaccine if recommended by the employer than by their doctors. This is probably a reflection of the current economic instability, and the individuals' wish to secure their employment. Among the employed people, employers can play a crucial role in overcoming COVID-19 vaccine hesitancy. Confidence and trust in vaccines among the workplaces and co-workers can lead to more individuals being vaccinated. Additionally, employers can plan ways to address potential barriers to getting the COVID-19 vaccine among the employers, create awareness, educate and share key tailored messages to build trust among the employees [36].

Our study showed that individuals with no education and lower-income were more likely to have COVID-19 vaccine hesitancy, these may be the same subgroup with lower educational attainment. More effort should be put into increasing awareness, knowledge, and addressing the myths and misinformation about COVID-19 infection and safety of COVID-19 vaccine among the population with no or lower educational achievements. The focus should be on using non-medical terminologies that can help in accurate and easy understanding of the

**Table 6. Multivariate socio-demographic predictors of responding "not sure" or "no" regarding acceptance to get vaccinated with COVID-19 vaccine.**

| Variables | Attributes | COVID-19 vaccine acceptance: Not Sure V/s Yes OR (95% CI) | COVID-19 vaccine acceptance: No V/s Yes OR (95% CI) |
|---|---|---|---|
| Region | Urban | **0.699(0.55–0.888)** | 0.907(0.748–1.101) |
| | Rural | Ref | |
| Age (in years) | 18–24 | **0.549(0.309–0.977)** | 1.019(0.642–1.615) |
| | 25–54 | 0.988(0.614–1.591) | **1.601(1.086–2.359)** |
| | 55–64 | 1.123(0.697–1.811) | 1.273(0.853–1.9) |
| | 65+ | Ref | |
| Gender | Male | 1.059(0.832–1.349) | 1.205(0.988–1.47) |
| | Female | Ref | |
| Education Level | No education | **2.799(1.103–7.108)** | **3.992(2.08–7.659)** |
| | Primary school | 2.277(0.864–6.001) | **4.8(2.448–9.412)** |
| | Middle School | 1.95(0.758–5.018) | **3.95(2.084–7.484)** |
| | High school | 1.628(0.654–4.051) | **2.998(1.625–5.533)** |
| | Intermediate or diploma | 0.901(0.353–2.302) | **2.546(1.375–4.714)** |
| | Graduate | 1.381(0.554–3.445) | 1.474(0.804–2.703) |
| | Profession/Honours | Ref | |
| Monthly Income (INR) | ≤10,001 | **3.632(1.649–8.001)** | 1.072(0.715–1.607) |
| | 10,002–29,972 | **10.299(4.879–21.741)** | **2.628(1.777–3.887)** |
| | 29,973–74,755 | **10(4.962–20.151)** | 1.373(0.972–1.939) |
| | 74,756–99,930 | **8.555(4.272–17.132)** | **0.636(0.443–0.913)** |
| | 99,931–199,861 | **2.896(1.304–6.429)** | 1.206(0.826–1.759) |
| | ≥199,862 | Ref | |
| Marital Status | Unmarried | 1.138(0.858–1.51) | 0.806(0.632–1.028) |
| | Others | 0.932(0.544–1.597) | 1.442(0.959–2.168) |
| | Married | Ref | |
| Employment Status | Others | 0.837(0.609–1.15) | **1.351(1.06–1.722)** |
| | Homemaker | 0.925(0.475–1.803) | 1.391(0.843–2.294) |
| | Employed | Ref | |

[a]OR: Odds ratio

[b]CI: Confidence interval

Significance taken at p < 0.05 (In Bold)

vaccine and infection-related educational messages. Additionally, individuals who have well-educated might have the ability to acquire accurate knowledge [37] as well as question any rumors prevailing about the COVID-19 vaccine by verifying from different sources. It would be important to assess the role of health literacy among individuals with higher education as a factor of enhanced COVID-19 vaccine acceptance. It is striking that the lower-income groups were hesitant or refused the COVID-19 vaccine even though it was available free of cost. The significant difference in COVID-19 vaccine acceptance among individuals belonging to lower and higher income groups is a cause of concern and demonstrates the need to bridge the gap of COVID-19 vaccine acceptance between the groups. Factors such as overcrowded and unsanitary living conditions, low immunity, higher likelihood of working outside the home, using public transport, and inability to follow social distancing at home or workplaces can put individuals in the low-income groups at an increased risk of getting COVID-19 infection [38]. The results of our study are similar to other studies showing lower education [23, 39–41] and low-income levels [8, 20, 32] associated with poor acceptance of the COVID-19 vaccine.

Additionally, our study demonstrated that less-educated individuals, having low income, and unemployed, retired, or lost jobs due to COVID-19 were more likely to refuse the COVID-19 vaccine.

Prior studies done on the factors influencing the COVID-19 vaccine found demographic factors such as age [39, 42] and gender [19, 23, 24] were significant predictors of acceptance of the COVID-19 vaccine. But in our study, we did not find old age or males to be significant predictors of COVID-19 vaccine acceptance demonstrating that the association between age, gender, and vaccine hesitancy is context-specific. The findings of our study demonstrate that living in an urban environment and age between 18 to 25 years is associated with a lower likelihood of COVID-19 vaccine hesitancy. Exposure to media and either themselves or being in contact with more educated individuals may play a role.

Results of our study demonstrate the need for tailored, data-driven, evidence-based transparent communication by various trusted sources of information to increase public trust in the recently developed COVID-19 vaccine, and incentivize individuals to get vaccinated. Health promotion is necessary especially among those who expressed intention to be vaccinated to be fully confident to accept the vaccine. In this study, about one-fourth of the respondents expressed probable acceptance but were a bit hesitant. Therefore, vaccination campaigns should focus on translating the high levels of probable acceptance into actual acceptance.

The study has some limitations. Firstly, it is cross-sectional, limited to one geographic setting, and difficult to assess causal and effect relationships. The pilot survey was done when the vaccination drive was just started for the general population in India. Hence, as more data, evidence, and information become accessible on the safety and effectiveness of COVID-19 vaccines, individuals might change their opinion towards vaccination. Despite the above limitations, our pilot study is among the few, to assess the urban-rural differences in COVID-19 vaccine acceptance in the Indian setting. The influence of region of residence on vaccine acceptance has been reported in numerous studies [33, 43] Since the pilot study was conducted in the initial days of vaccine rollout for the general population our data can be used as baseline data to future assess the effectiveness of the interventions focused on enhancing the uptake of the vaccines.

## Conclusion

Reasons for hesitancy are different from person to person and therefore, a one-size-fits-all approach cannot be utilized to implement an effective intervention addressing vaccine hesitancy. To minimize the public health threat of the COVID-19 pandemic in India, a significant proportion of the population needs to be vaccinated. It is important to address vaccine hesitancy in rural parts of India along with aggressive campaigns in urban parts of India. Unvaccinated individuals from rural areas moving back to urban areas to work or vis-versa may lead to new strains and increase urban-rural transmission of infection. Furthermore, a potential fourth COVID-19 pandemic wave may target unaffected and unvaccinated sections of the population. Hence, it is significant to spread awareness about the COVID-19 vaccine in zones with inadequate healthcare infrastructure and among those people who are prone to COVID-19 vaccine hesitancy.

## Supporting information

**S1 Appendix. Survey questionnaire.**
(DOCX)

**S2 Appendix. Informed consent form.**
(PDF)

**S3 Appendix. Minimal and anonymized data set underlying the results.**
(XLSX)

**S4 Appendix. Health status of the respondents.**
(DOCX)

**S5 Appendix. Distribution of health status against acceptance of COVID-19 vaccine.**
(DOCX)

## Author Contributions

**Conceptualization:** Ashish Joshi.

**Formal analysis:** Ashish Joshi, Ashruti Bhatt.

**Investigation:** Ashish Joshi, Krishna Mohan Surapaneni.

**Methodology:** Ashish Joshi, Mahima Kaur.

**Project administration:** Ashish Joshi, Krishna Mohan Surapaneni.

**Resources:** Ashish Joshi, Krishna Mohan Surapaneni.

**Software:** Ashish Joshi.

**Supervision:** Ashish Joshi, Krishna Mohan Surapaneni.

**Validation:** Ashish Joshi, Mahima Kaur.

**Visualization:** Ashish Joshi, Mahima Kaur.

**Writing – original draft:** Ashish Joshi, Mahima Kaur.

**Writing – review & editing:** Ashish Joshi, Krishna Mohan Surapaneni, Mahima Kaur, Ashruti Bhatt, Denis Nash, Ayman El-Mohandes.

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
