## [Decision Letter · Decision Letter 0]

15 Dec 2021

PONE-D-21-35060A cross sectional study to examine factors influencing COVID-19 vaccine acceptance, hesitancy and refusal in urban and rural settings in Tamil Nadu, IndiaPLOS ONE

Dear Dr. Joshi,

Thank you for submitting your manuscript to PLOS ONE. After careful consideration, we feel that it has merit but does not fully meet PLOS ONE’s publication criteria as it currently stands. Therefore, we invite you to submit a revised version of the manuscript that addresses the points raised during the review process.

ACADEMIC EDITOR: As the study has employed a big sample size and it has been found that the tables and the sample (n) does not match. Kindly clarify on similar terms. Also, provide ethical clearance no from the institution in the revised manuscript. Such issues would be dealt seriously in the revised version.

We look forward to receiving your revised manuscript.

Kind regards,

Sheikh Mohd Saleem, MBBS, MD

Academic Editor

PLOS ONE

Journal Requirements:

3. Please amend your current ethics statement to address the following concerns:

a) Did participants provide their written or verbal informed consent to participate in this study?

Reviewers' comments:

Reviewer's Responses to Questions

**Comments to the Author**

1. Is the manuscript technically sound, and do the data support the conclusions?

Reviewer #1: Partly

Reviewer #2: Yes

2. Has the statistical analysis been performed appropriately and rigorously? 

Reviewer #1: Yes

Reviewer #2: Yes

3. Have the authors made all data underlying the findings in their manuscript fully available?

Reviewer #1: No

Reviewer #2: Yes

4. Is the manuscript presented in an intelligible fashion and written in standard English?

Reviewer #1: Yes

Reviewer #2: Yes

5. Review Comments to the Author

Reviewer #1: 1. The study has been carried out in a rural area in Tamil Nadu. Shouldn't the corresponding author be from the institute where the data has been collected?

2. The study has been carried out in government PHC's across Tamil Nadu, however there has been no mention of permission being granted/ ethical clearance frim the government to carry out this study. Kindly clarify.

3. Were pregnant and lactating women included in the study? If they were, it would have contributed to more hesitancy. At this time, vaccines were still not recommended for pregnant or lactating women.

4. The sample size is 3130. However, all tables that have been included in the study do not have 'n' specified in the heading and the numbers in the cells don't add up to 3130. Kindly clarify. (Table 1-4)

5. This study is very extensive and has collected a lot of data. However, I think there is too much data for one paper. The emphasis should be on vaccine naiive individuals with few causes of hesitancy being studied.

6. Patients who were vaccinated with the COVID vaccine before and questions about cost and effectiveness are best left to another paper. Table 2, 7 and table 4 do not contribute valuable data.

7. The numbers in Table 5a do not add up to 3130. Table 5 b can be added to supplementary data or summarized into one line at the end.

8. This paper has a good concept but has too much data. Was a focus group discussion held prior to the study to determine the main causes of vaccine hesitancy? That would help guide us better on what variables to concentrate on.

9. Discussion is too long. Kindly shorten it to be shorter and clearer.

Reviewer #2: 1) Fragmentation of sentence needed in line 74-75.

2) Lines 86-93 are misplaced. It would be wise to elaborate on it in the "Discussion".

3) Line 99-100- Check for grammar.

4) Lines 130-152 are misplaced and irrelevant.

5) Choose only one format to discuss the data in lines 298-304. (e.g.- either "10%" or "ten percent")

6) Repetitive use of words that convey the same message ("efficacy" and "effectiveness") in lines 299-301 should be avoided. Kindly re-phrase the sentence.

6. PLOS authors have the option to publish the peer review history of their article (what does this mean?). If published, this will include your full peer review and any attached files.

Reviewer #1: No

Reviewer #2: **Yes: **Aditya Anurag

---

## [Author Response · Author response to Decision Letter 0]

28 Jan 2022

Reviewer #1: 

1. The study has been carried out in a rural area in Tamil Nadu. Shouldn't the corresponding author be from the institute where the data has been collected?

Response: The study has been carried out in both rural and urban settings. The corresponding author is responsible for the conceptualization of the study, study design, research questions and the institute in Tamil Nadu is responsible for the implementation of the study protocol and the data collection. 

2. The study has been carried out in government PHC's across Tamil Nadu, however there has been no mention of permission being granted/ ethical clearance from the government to carry out this study. Kindly clarify.

Response: Yes, the study was carried out after permission was granted by the PHCs. 

3. Were pregnant and lactating women included in the study? If they were, it would have contributed to more hesitancy. At this time, vaccines were still not recommended for pregnant or lactating women.

Response: We thank the reviewer for identifying the missing component in the exclusion criteria. The pregnant and lactating women were excluded from the study. The exclusion criteria have been revised accordingly. 

4. The sample size is 3130. However, all tables that have been included in the study do not have 'n' specified in the heading and the numbers in the cells don't add up to 3130. Kindly clarify. (Table 1-4)

Response: We appreciate the feedback from the reviewer. As suggested, we have specified the ‘n’ in each heading of the table. Since, the data had missing variables the it was analysed separately for every variable. Therefore, the total sample size does not match the ‘n’ for all the variables included in the study. 

5. This study is very extensive and has collected a lot of data. However, I think there is too much data for one paper. The emphasis should be on vaccine naiive individuals with few causes of hesitancy being studied.

Response: As suggested in the following comment, we have deleted table 7 and 8 to shorten the paper so that it contains valuable data. Table 2 has been added as a supplementary material. 

6. Patients who were vaccinated with the COVID vaccine before and questions about cost and effectiveness are best left to another paper. Table 2, 7 and table 4 do not contribute valuable data.

Response: As suggested in the following comment, we have deleted table 7 and 8 to shorten the paper so that it contains valuable data. Table 2 has been added as a supplementary material. Table 4 talks about the vaccine acceptance and hesitancy hence we have not deleted the table. 

7. The numbers in Table 5a do not add up to 3130. Table 5 b can be added to supplementary data or summarized into one line at the end.

Response: Since, the data had missing variables the it was analysed separately for every variable. As suggested, we have added the Table5b to the supplementary data. 

8. This paper has a good concept but has too much data. Was a focus group discussion held prior to the study to determine the main causes of vaccine hesitancy? That would help guide us better on what variables to concentrate on.

Response: We thank the reviewer for the feedback. But we did not hold a focus group discussion prior to the study. Only open-ended questions were asked to determine the main causes of vaccine hesitancy among the population. 

9. Discussion is too long. Kindly shorten it to be shorter and clearer.

Response: As suggested, we have shortened the discussion. Additionally, we have divided the discussion into conclusion as well to clearly summarize the key message. 

Reviewer #2: 

1) Fragmentation of sentence needed in line 74-75.

Response: As suggested, we have fragmented the sentence as below:

“Concerns about vaccine hesitancy are growing globally, especially in populous countries with low literacy. This prompted the WHO to declare it among the top ten health threats in 2019.”

2) Lines 86-93 are misplaced. It would be wise to elaborate on it in the "Discussion".

Response: We appreciate the reviewer’s comment. But, the lines were added to state the prevalence of the vaccine hesitancy rates across the world so as it give an overview of the studies conducted in this domain. Besides, we have mentioned and elaborated the prevalence of the vaccine hesitancy in the “Discussion” as well.

3) Line 99-100- Check for grammar.

Response: As suggested by the reviewer, we have corrected the lines to: 

“The emergence of contagious and/or virulent variants, lower immunity of the vulnerable individuals, and lack of participation in COVID-19 preventive measures may have contributed to a resurgence of the COVID-19 cases”. 

4) Lines 130-152 are misplaced and irrelevant.

Response: We thank the reviewer for the comment. The lines were added to provide a brief background of the COVID-19 infection and the vaccination rates in the state to strengthen the rationale behind conducting the cross-sectional study. 

5) Choose only one format to discuss the data in lines 298-304. (e.g.- either "10%" or "ten percent")

Response: As suggested we have edited the format of the data to maintain the consistency. 

6) Repetitive use of words that convey the same message ("efficacy" and "effectiveness") in lines 299-301 should be avoided. Kindly re-phrase the sentence.

Response: As suggested the lines have been re-phrased.

---

## [Decision Letter · Decision Letter 1]

4 Mar 2022

PONE-D-21-35060R1A cross sectional study to examine factors influencing COVID-19 vaccine acceptance, hesitancy and refusal in urban and rural settings in Tamil Nadu, IndiaPLOS ONE

Dear Dr. Joshi,

Thank you for submitting your manuscript to PLOS ONE. After careful consideration, we feel that it has merit but does not fully meet PLOS ONE’s publication criteria as it currently stands. Therefore, we invite you to submit a revised version of the manuscript that addresses the points raised during the review process.

ACADEMIC EDITOR: Please insert comments here and delete this placeholder text when finished. Be sure to:

Kindly address each comment from the reviewer in the next revision which you shall submit.

We look forward to receiving your revised manuscript.

Kind regards,

Sheikh Mohd Saleem, MBBS, MD

Academic Editor

PLOS ONE

Additional Editor Comments (if provided):

The article still needs some modifications and revision. Kindly address each comment of the reviews seriously. A major revision is warranted.

Reviewers' comments:

Reviewer's Responses to Questions

**Comments to the Author**

1. If the authors have adequately addressed your comments raised in a previous round of review and you feel that this manuscript is now acceptable for publication, you may indicate that here to bypass the “Comments to the Author” section, enter your conflict of interest statement in the “Confidential to Editor” section, and submit your "Accept" recommendation.

Reviewer #1: All comments have been addressed

Reviewer #2: All comments have been addressed

2. Is the manuscript technically sound, and do the data support the conclusions?

Reviewer #1: Yes

Reviewer #2: Yes

3. Has the statistical analysis been performed appropriately and rigorously? 

Reviewer #1: Yes

Reviewer #2: I Don't Know

4. Have the authors made all data underlying the findings in their manuscript fully available?

Reviewer #1: Yes

Reviewer #2: Yes

5. Is the manuscript presented in an intelligible fashion and written in standard English?

Reviewer #1: Yes

Reviewer #2: Yes

6. Review Comments to the Author

Reviewer #1: Thank you for the changes that have been made.

I still feel the article is long and certain points are being repeated in the results section and discussion section.

1. Line 382-384 and 399-402 can be removed. This has already been explained in the comments section.

2. Line 494 comments on a 'third' pandemic wave. Consider changing it to fourth, since we are at the tail end of the third wave in India.

3. To make the article comprehensive and easier to read, I would suggest shortening the text explanation before each table. The same information is again listed out in the discussion section.

4. Ethical clearance has been obtained from the institution. However, urban and rural PHC's are not run by the institution (Panimalar) but are run by the State Government of Tamil Nadu. Would a separate ethical clearance be required from the State government to utilize data of patients attending their PHC's?

Reviewer #2: The manuscript is still a bit lengthy and sometimes wavers from the topic. Lines 110-120 in Introduction and lines 144-155 in Materials and Methods may be removed/cut-short as they seem inappropriately placed and are making the manuscript unnecessarily lengthy.

7. PLOS authors have the option to publish the peer review history of their article (what does this mean?). If published, this will include your full peer review and any attached files.

Reviewer #1: No

Reviewer #2: **Yes: **ADITYA ANURAG

---

## [Author Response · Author response to Decision Letter 1]

3 May 2022

Reviewer #1: Thank you for the changes that have been made.

I still feel the article is long and certain points are being repeated in the results section and discussion section.

1. Line 382-384 and 399-402 can be removed. This has already been explained in the comments section.

Response: We have deleted the suggested lines. 

2. Line 494 comments on a 'third' pandemic wave. Consider changing it to fourth, since we are at the tail end of the third wave in India.

Response: Thank you for the feedback. We can changed the third wave to fourth COVID-19 pandemic wave. 

3. To make the article comprehensive and easier to read, I would suggest shortening the text explanation before each table. The same information is again listed out in the discussion section.

Response: We thank the reviewer for the feedback. We have shorten the text explanation before each table and deleted the few lines listed again in the discussion. We have listed only the key findings in the text explanation before each table. 

4. Ethical clearance has been obtained from the institution. However, urban and rural PHC's are not run by the institution (Panimalar) but are run by the State Government of Tamil Nadu. Would a separate ethical clearance be required from the State government to utilize data of patients attending their PHC's?

Response: The Institutional Ethics Committee of Panimalar Medical College Hospital & Research Institute (PMCHRI-IHEC) has been approved and recognized by the CDSCO, NECRBHR, Department of Health Research, Government of India. The PMCHRI-IHEC is an independent authority to approve the clinical trials or any protocol involving the human subjects as study participants. Once the protocol is approved by the IHEC, the study can be carried out as per the approved protocol without any deviation across the state as mentioned in the study protocol and as approved by the IHEC. The letter stating the same has been uploaded in the supporting document. 

Reviewer #2: The manuscript is still a bit lengthy and sometimes wavers from the topic. Lines 110-120 in Introduction and lines 144-155 in Materials and Methods may be removed/cut-short as they seem inappropriately placed and are making the manuscript unnecessarily lengthy.

Response: Thank you for the feedback. We have removed the suggested lines (110-120) to make the article comprehensive. We have cut-short the lines 144-155. We have also deleted the lines 133-137 to shorten the manuscript.

---

## [Decision Letter · Decision Letter 2]

19 May 2022

A cross sectional study to examine factors influencing COVID-19 vaccine acceptance, hesitancy and refusal in urban and rural settings in Tamil Nadu, India

PONE-D-21-35060R2

Dear Dr. Joshi,

We’re pleased to inform you that your manuscript has been judged scientifically suitable for publication and will be formally accepted for publication once it meets all outstanding technical requirements.

Kind regards,

Sheikh Mohd Saleem, MBBS, MD

Academic Editor

PLOS ONE

Additional Editor Comments (optional):

Reviewers' comments:

Reviewer's Responses to Questions

**Comments to the Author**

1. If the authors have adequately addressed your comments raised in a previous round of review and you feel that this manuscript is now acceptable for publication, you may indicate that here to bypass the “Comments to the Author” section, enter your conflict of interest statement in the “Confidential to Editor” section, and submit your "Accept" recommendation.

Reviewer #1: All comments have been addressed

2. Is the manuscript technically sound, and do the data support the conclusions?

Reviewer #1: Yes

3. Has the statistical analysis been performed appropriately and rigorously? 

Reviewer #1: Yes

4. Have the authors made all data underlying the findings in their manuscript fully available?

Reviewer #1: Yes

5. Is the manuscript presented in an intelligible fashion and written in standard English?

Reviewer #1: Yes

6. Review Comments to the Author

Reviewer #1: All questions have been addressed. Introduction can be shortened if possible.

7. PLOS authors have the option to publish the peer review history of their article (what does this mean?). If published, this will include your full peer review and any attached files.

Reviewer #1: No

---

## [Editor Report · Acceptance letter]

31 May 2022

PONE-D-21-35060R2 

A cross sectional study to examine factors influencing COVID-19 vaccine acceptance, hesitancy and refusal in urban and rural settings in Tamil Nadu, India 

Dear Dr. Joshi:

I'm pleased to inform you that your manuscript has been deemed suitable for publication in PLOS ONE. Congratulations! Your manuscript is now with our production department. 

Kind regards, 

on behalf of

Dr. Sheikh Mohd Saleem 

Academic Editor

PLOS ONE